# Space station biomining experiment demonstrates rare earth element extraction in microgravity and Mars gravity

Charles S. Cockell [1,11✉], Rosa Santomartino [1,11], Kai Finster [2], Annemiek C. Waajen [1], Lorna J. Eades[3], Ralf Moeller[4], Petra Rettberg [4], Felix M. Fuchs[4,5], Rob Van Houdt [6], Natalie Leys [6], Ilse Coninx[6], Jason Hatton[7], Luca Parmitano[7], Jutta Krause[7], Andrea Koehler[7], Nicol Caplin[7], Lobke Zuijderduijn[7], Alessandro Mariani[8], Stefano S. Pellari[8], Fabrizio Carubia[8], Giacomo Luciani[8], Michele Balsamo[8], Valfredo Zolesi[8], Natasha Nicholson[1], Claire-Marie Loudon[1], Jeannine Doswald-Winkler[9], Magdalena Herová[9], Bernd Rattenbacher[9], Jennifer Wadsworth[10], R. Craig Everroad[10] & René Demets[7]

Microorganisms are employed to mine economically important elements from rocks, including the rare earth elements (REEs), used in electronic industries and alloy production. We carried out a mining experiment on the International Space Station to test hypotheses on the bioleaching of REEs from basaltic rock in microgravity and simulated Mars and Earth gravities using three microorganisms and a purposely designed biomining reactor. *Sphingomonas desiccabilis* enhanced mean leached concentrations of REEs compared to non-biological controls in all gravity conditions. No significant difference in final yields was observed between gravity conditions, showing the efficacy of the process under different gravity regimens. *Bacillus subtilis* exhibited a reduction in bioleaching efficacy and *Cupriavidus metallidurans* showed no difference compared to non-biological controls, showing the microbial specificity of the process, as on Earth. These data demonstrate the potential for space biomining and the principles of a reactor to advance human industry and mining beyond Earth.

[1] UK Centre for Astrobiology, School of Physics and Astronomy, University of Edinburgh, Edinburgh, UK. [2] Department of Bioscience–Microbiology, Ny Munkegade 116, Building 1540, 129, 8000 Aarhus C, Denmark. [3] School of Chemistry, University of Edinburgh, Edinburgh, UK. [4] Radiation Biology Department, German Aerospace Center (DLR), Institute of Aerospace Medicine, Linder Hoehe, Köln, Germany. [5] Institute of Electrical Engineering and Plasma Technology, Faculty of Electrical Engineering and Information Sciences, Ruhr University Bochum, Bochum, Germany. [6] Microbiology Unit, Belgian Nuclear Research Centre, SCK CEN, Mol, Belgium. [7] ESTEC, Keplerlaan 1, 2201 AZ Noordwijk, Netherlands. [8] Kayser Italia S.r.l., Via di Popogna, 501, 57128 Livorno, Italy. [9] BIOTESC, Hochschule Luzern Technik & Architektur, Lucerne School of Engineering and Architecture, Obermattweg 9, 6052 Hergiswil, Switzerland. [10] Exobiology Branch, NASA Ames Research Center, Moffett Field, CA, USA. [11] These authors contributed equally: Charles S. Cockell, Rosa Santomartino. ✉email: c.s.cockell@ed.ac.uk

On Earth, microorganisms play prominent roles in natural processes such as the weathering of rocks into soils and the cycling of elements in the biosphere. Microorganisms are also used in diverse industrial and manufacturing processes[1–4], for example in the process called biomining (or bioleaching)[5,6]. Microorganisms can catalyse the extraction of valuable elements from rocks, such as copper and gold[7,8]. This process can in some circumstances reduce the environmentally damaging use of toxic compounds such as cyanides[9,10]. These microbial interactions with minerals are also used to decontaminate polluted soils, in a process called bioremediation[10]. Acidophilic iron and sulfur-oxidisers are often used to biomine economic elements from sulfidic ores, but heterotrophic microorganisms, including bacteria and fungi, can be effective in bioleaching in environments with circumneutral or alkaline pH. These organisms can enable leaching by changing the local pH in the environment, for example by the release of protons or organic acids. Alternatively, leaching and sequestration of elements can occur as a consequence of the release of complexing compounds[11–15].

Of important economic and practical interest are rare earth elements (REEs), which include the lanthanides, scandium and yttrium. On account of their physical properties, including ferromagnetism and luminescence, REEs are used in electronic devices such as cell phones and computer screens, as well as in catalysis, metal alloy and magnet production, and many other high-technology applications. Some REEs are identified as short-term near-critical elements[16], meaning that the demand will soon outstrip supply. Microorganisms are known to be able to mobilise REEs. For example, REEs are used as a cofactor in alcohol dehydrogenases in diverse microbial taxa[17,18], and they were shown to be essential for the survival of an acidophilic methanotroph in a volcanic mudpot[19]. The ability of microorganisms to mobilise REEs from rocks has been shown for a variety of different mineral matrices[20,21].

As humans explore and potentially settle in space, microbe–mineral interactions have been recognised to be important, including in biomining[22–24]. In addition to mining beyond the Earth, advancing our understanding of microbe–mineral interactions in space could be applied to: (1) soil formation from nutrient-poor rocks[22], (2) formation of biocrusts to control dust and surface material in enclosed pressurised spaces[25], (3) use of regolith as feedstock within microbial segments of life support systems[26], (4) use of regolith and microbes in microbial fuel cells (biofuel)[22], (5) biological production of mineral construction materials[27]. All of these diverse applications have in common that they require experimental investigations on how microbes attach to, and interact with, rock and regolith materials in space environments. Furthermore, there is a need to know how organisms alter ion leaching and mineral degradation in altered gravity regimens, which will occur in any extraterrestrial location.

Altered gravity conditions, such as microgravity, are known to influence microbial growth and metabolic processes[28–30]. Although the capacity of prokaryotes to directly sense gravity remains a point of discussion, gravity influences sedimentation and convection in bulk fluids[31]. By allowing for thermal convection and sedimentation, gravity is thought to affect the mixing of nutrients and waste, thereby influencing microbial growth and metabolism[32–35]. Based on these considerations, we hypothesised that altered gravity regimens would induce changes in microbial interactions with minerals, and thus bioleaching.

In this work, we present the results of the European Space Agency BioRock experiment, performed on the International Space Station (ISS) in 2019 to investigate the leaching of elements from basalt[36–38], an analogue for much of the regolith material on the Moon and Mars, by three species of heterotrophic microorganisms. The experiment compared bioleaching at three different levels of gravity: microgravity, simulated Mars and terrestrial gravity. Results are reported on the bioleaching of REEs, demonstrating the effective use of microorganisms in biomining beyond Earth using a miniaturised space biomining reactor.

## Results

**REE biomining in space.** Data were acquired using the BioRock biomining reactor, designed for these experiments (Fig. 1) which contained basaltic rock with known REE composition (Table 1) and major elements (Supplementary Table 1). REEs bioleached into solution were measured for all three organisms (*S. desiccabilis*, *B. subtilis*, *C. metallidurans*) in all three gravity conditions (microgravity, simulated Mars and Earth gravity) and for non-biological controls (Fig. 2, Supplementary Fig. 1 and Supplementary Table 2). The concentrations of leached REEs in biological and non-biological condition generally followed the trends expected from their abundance in the basaltic rock (Table 1; Supplementary Table 2). Elements with the highest abundance (*e.g.* Ce and Nd) showed the highest leached concentrations while elements with lowest abundance (Tb, Tm and Lu) exhibited the lowest concentrations.

Statistical analysis across all three organisms and the three gravity conditions tested in space showed a significant effect of the organism (ANOVA: $F_{(2,369)} = 87.84$, $p = 0.001$) on bioleaching. Post-hoc Tukey tests showed all pairwise comparisons between organisms to be significant ($p < 0.001$). There was a non-significant effect when gravity conditions were compared (ANOVA: $F_{(2,369)} = 0.202$, $p = 0.818$). The interaction between gravity and the organism was not significant (ANOVA: $F_{(4, 369)} = 1.75$, $p = 0.138$).

Statistical analysis was carried out on *S. desiccabilis* bioleaching. Comparing the difference between biological samples and the non-biological controls in each gravity condition for *S. desiccabilis* showed that microgravity was not significant (ANOVA: $F_{(1,69)} = 2.43$, $p = 0.124$), but significant differences between the biological experiments and the non-biological controls were observed in simulated Mars (ANOVA: $F_{(1,83)} = 14.14$, $p < 0.0001$) and Earth gravity (ANOVA: $F_{(1,83)} = 24.20$, $p < 0.0001$). The difference in bioleaching between gravity conditions was not significant (ANOVA: $F_{(2,123)} = 1.60$, $p = 0.206$) for *S. desiccabilis*.

For *S. desiccabilis*, across all individual REEs and across all three gravity conditions on the ISS, the organism had leached 111.9% to 429.2% of the non-biological controls (Fig. 3a and Supplementary Table 3). Student's $t$ tests were used to examine the concentration of individual REEs bioleached compared to non-biological controls. Bioleaching was significantly higher than non-biological controls under simulated Mars and Earth gravity for individual REEs ($p < 0.05$, Student's $t$ test, Supplementary Table 4), except for Pr and Nd which were significantly higher at the $p < 0.1$ level, and not significant for Ce in simulated Mars gravity ($p = 0.102$). For the microgravity condition, none of individual REE concentrations in the biological experiment was significantly higher than the non-biological control ($p > 0.05$) (Supplementary Table 4). The standard deviations of the microgravity biological and non-biological controls for the individual REEs for *S. desiccabilis* were, apart from Pr in the biological experiment, higher than for *B. subtilis* and *C. metallidurans*.

Student's $t$ test comparisons were carried out between the concentrations of bioleached REEs in different gravities for each element for *S. desiccabilis* (Supplementary Table 4). Comparison between the simulated Mars gravity and simulated Earth gravity

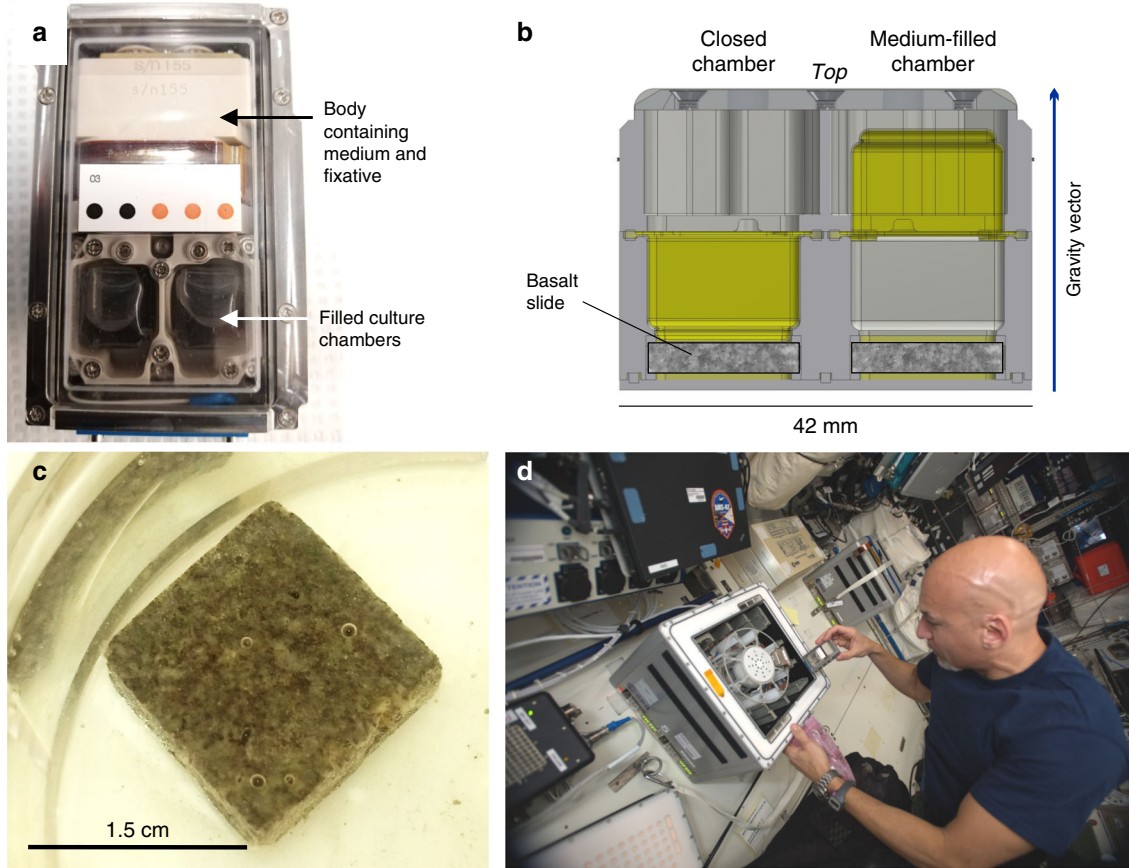

**Fig. 1 The BioRock Experimental Unit. a** Top-down image of one Experimental Container (EC) containing one EU (Experimental Unit) showing both culture chambers inflated with medium. **b** Sideways cross section through culture chamber showing location of basalt slide at the back of the chamber and principle of medium injection and inversion of membrane (shown here in yellow; left side closed, right side inflated with medium). **c** Image of basalt slide in a Petri dish submerged in 50% R2A in a ground experiment. **d** ESA astronaut Luca Parmitano inserts an EC into a KUBIK incubator on board the International Space Station (image credit to ESA).

**Table 1 Content of rare earth elements (REEs; reported as µg/g; mean ± standard deviation) in the basalt substrate used in this experiment and concentrations (total nanograms leached into the chamber fluid volume of 6 mL) at the end of the BioRock experiment in *S. desiccabilis* bioleaching chambers and non-biological controls on-board the International Space Station.**

| | | *S. desiccabilis* | | | non-biological control | | |
|---|---|---|---|---|---|---|---|
| **REE** | ***Concentration in basalt* (µg/g)** | **Microgravity** | **Mars gravity** | **Earth gravity** | **Microgravity** | **Mars gravity** | **Earth gravity** |
| La | 6.81 | 3.60 ± 1.26 | 4.96 ± 0.51 | 3.74 ± 0.51 | 3.22 ± 2.20 | 2.56 ± 0.89 | 1.66 ± 0.23 |
| Ce | 13.53 | 8.85 ± 2.89 | 9.26 ± 1.94 | 7.18 ± 0.99 | 6.45 ± 3.99 | 5.79 ± 2.06 | 4.39 ± 1.26 |
| Pr | 2.32 | 1.12 ± 0.43 | 1.67 ± 0.48 | 1.07 ± 0.11 | 0.96 ± 0.64 | 0.85 ± 0.28 | 0.48 ± 0.04 |
| Nd | 11.57 | 5.35 ± 2.02 | 7.89 ± 1.99 | 5.20 ± 0.47 | 4.68 ± 3.49 | 4.28 ± 1.46 | 2.28 ± 0.24 |
| Sm | 3.04 | 1.44 ± 0.57 | 2.03 ± 0.36 | 1.42 ± 0.12 | 1.13 ± 0.90 | 1.06 ± 0.37 | 0.54 ± 0.07 |
| Eu | 1.13 | 0.51 ± 0.16 | 0.66 ± 0.07 | 0.53 ± 0.04 | 0.44 ± 0.25 | 0.42 ± 0.11 | 0.27 ± 0.03 |
| Gd | 3.67 | 2.03 ± 0.86 | 2.93 ± 0.51 | 2.18 ± 0.13 | 1.60 ± 1.37 | 1.36 ± 0.52 | 0.70 ± 0.10 |
| Tb | 0.57 | 0.42 ± 0.14 | 0.57 ± 0.08 | 0.44 ± 0.01 | 0.30 ± 0.21 | 0.26 ± 0.07 | 0.16 ± 0.02 |
| Dy | 3.92 | 2.82 ± 1.00 | 3.99 ± 0.55 | 3.08 ± 0.21 | 1.86 ± 1.43 | 1.58 ± 0.52 | 0.92 ± 0.11 |
| Ho | 0.80 | 0.69 ± 0.27 | 0.98 ± 0.08 | 0.78 ± 0.08 | 0.45 ± 0.37 | 0.36 ± 0.13 | 0.20 ± 0.03 |
| Er | 2.44 | 2.34 ± 1.01 | 3.37 ± 0.22 | 2.75 ± 0.32 | 1.49 ± 1.26 | 1.17 ± 0.47 | 0.64 ± 0.11 |
| Tm | 0.29 | 0.42 ± 0.16 | 0.58 ± 0.04 | 0.49 ± 0.06 | 0.29 ± 0.19 | 0.24 ± 0.07 | 0.16 ± 0.01 |
| Yb | 2.11 | 2.44 ± 1.09 | 3.52 ± 0.36 | 2.83 ± 0.35 | 1.47 ± 1.19 | 1.16 ± 0.44 | 0.67 ± 0.11 |
| Lu | 0.31 | 0.49 ± 0.20 | 0.68 ± 0.08 | 0.57 ± 0.07 | 0.33 ± 0.22 | 0.27 ± 0.08 | 0.18 ± 0.02 |

($n = 3$ biologically independent samples with the exception of one non-biological microgravity and non-biological ground control sample which are not included. Full data set in Supplementary Table 2).

showed that the concentrations of five elements (La, Sm, Eu, Tb, Ho) were significantly different at the $p < 0.05$ level and five more elements (Gd, Dy, Er, Tm, Yb) at the $p < 0.1$ level, with simulated Earth gravity values being higher. These differences were more evident among the 'heavy' REEs (elements from Gd up to Lu) (Fig. 3a). The total quantity of REEs released by *S. desiccabilis* as a percentage of the available quantity in the basalt, ranged between $1.17 \times 10^{-1}$ and $2.41 \times 10^{-2}$% (Supplementary Table 5).

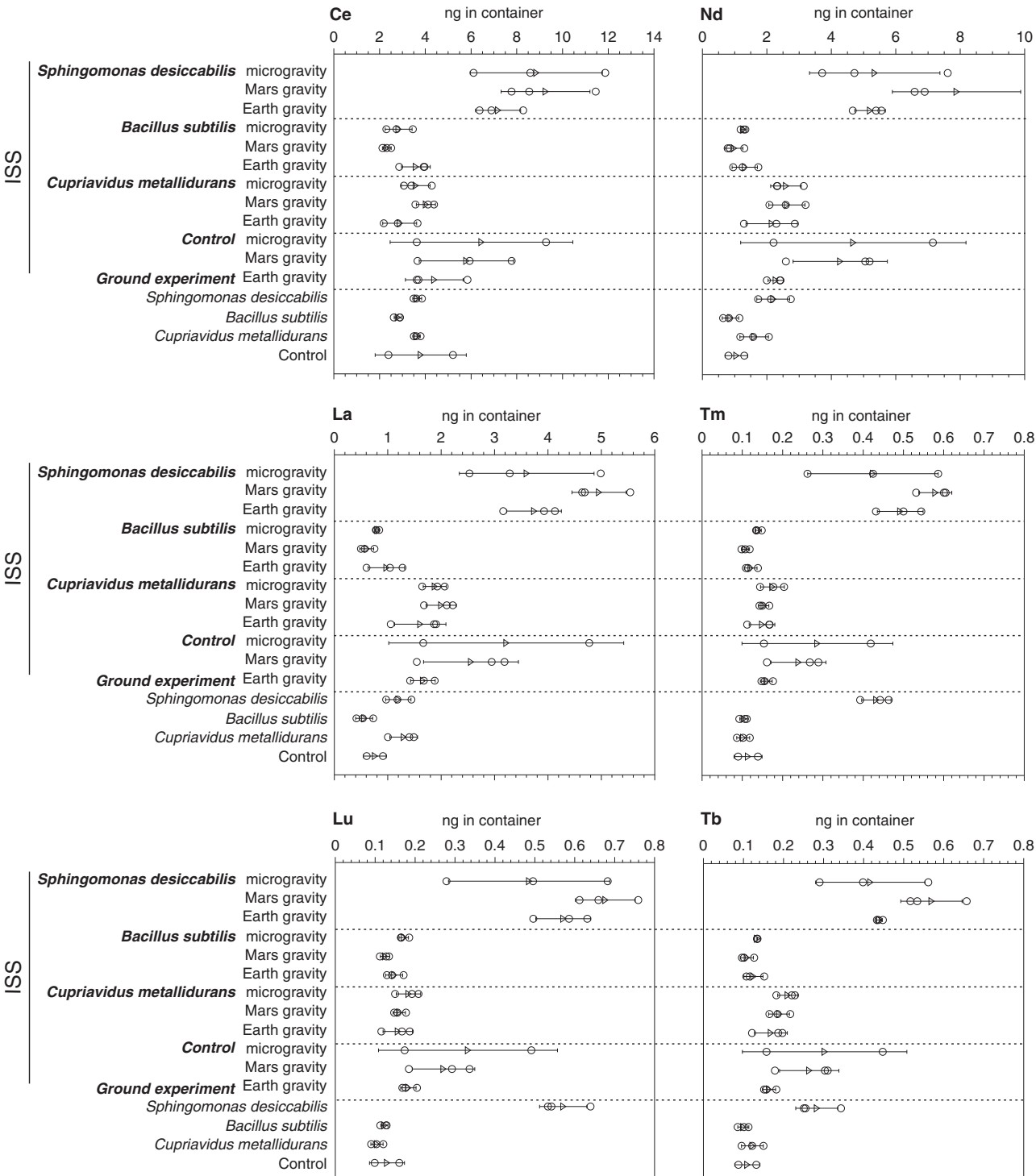

**Fig. 2 Bioleaching and control leaching of the most and least abundant rare earth elements.** Concentrations (ng in total chamber liquid) of rare earth elements (REEs) in each of the experimental flight and ground control samples at the end of the experiment (described in the text) for each of the three organisms and non-biological controls. The three most (Ce, Nd, La) and least (Tm, Lu, Tb) abundant REEs are shown here (all others in Supplemental Fig. 1). ISS shows the International Space Station flight experiments. Circles show triplicate measurements (n = 3 biologically independent samples. One non-biological microgravity and non-biological ground control sample were lost and are not shown) and the mean is given as a triangle. Error bars represent standard deviations.

Identical statistical analysis was carried out for bioleaching experiments with *B. subtilis* and *C. metallidurans*. For *B. subtilis*, the quantity of REEs bioleached was significantly less than the non-biological controls in microgravity (ANOVA: $F_{(1,69)} = 13.05$, $p < 0.001$) and simulated Mars gravity (ANOVA: $F_{(1,83)} = 29.55$, $p < 0.0001$), but marginally not significant in Earth gravity (ANOVA: $F_{(1,83)} = 3.79$, $p = 0.055$). The difference in the concentrations of REEs bioleached between gravity conditions was not significant (ANOVA: $F_{(2,123)} = 1.45$, $p = 0.240$).

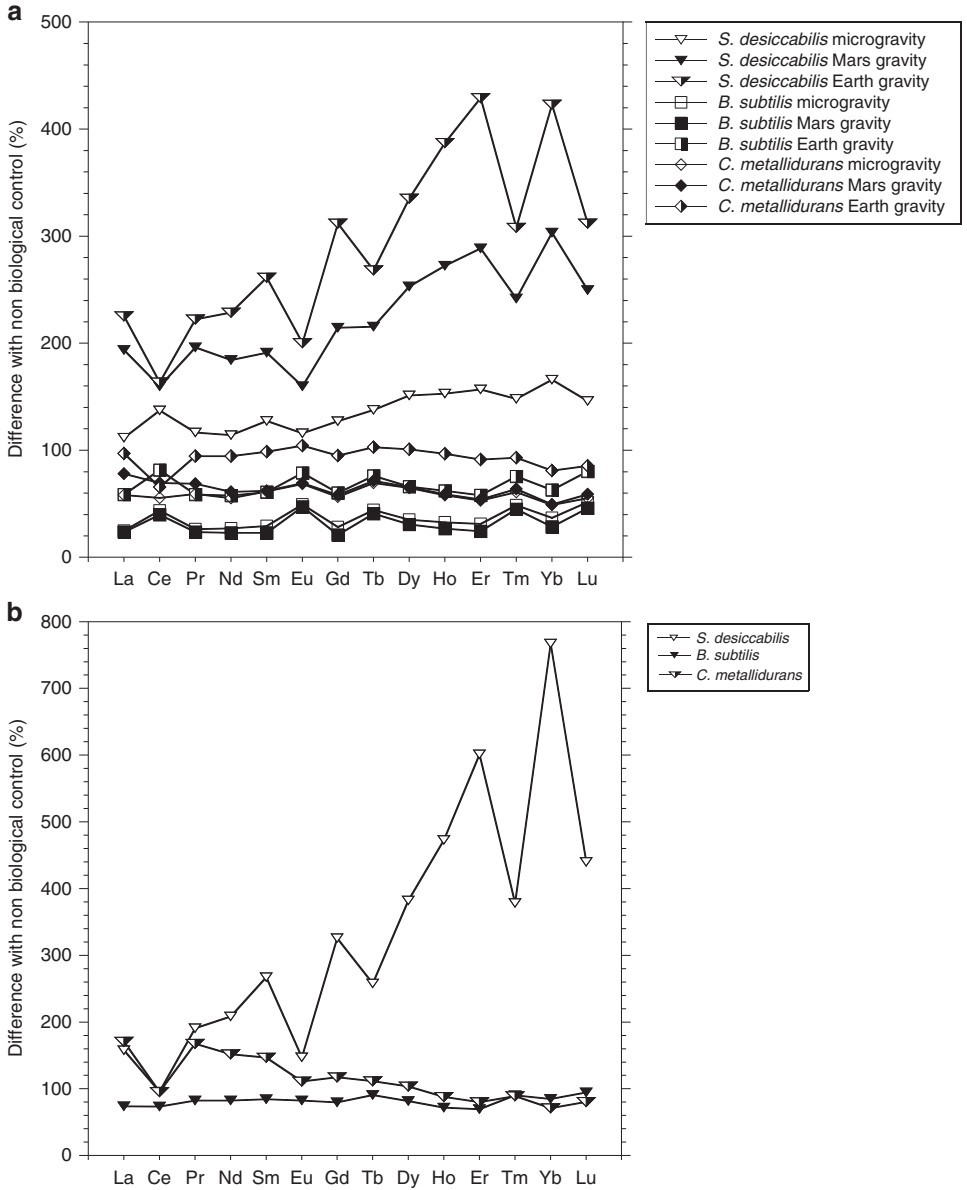

**Fig. 3 Effects of microorganisms on rare earth element leaching. a** Relative (%) difference in mean concentration of leached REEs in the bulk fluid between biological experiments and non-biological controls showing microgravity, simulated Mars and Earth gravities on the International Space Station for the three microorganisms. **b** Ground (true Earth gravity control) experiment for the three microorganisms. Standard deviations reported in Supplemental Table 3, statistics reported in the main text.

For *C. metallidurans*, the difference between bioleaching and the non-biological controls was not significant in all three gravity conditions: microgravity (ANOVA: $F(1,69) = 2.25$, $p < 0.138$), simulated Mars (ANOVA: $F(1,83) = 3.47$, $p < 0.066$), and Earth gravity (ANOVA: $F(1,83) = 0.265$, $p = 0.608$). The difference in bioleaching between gravity conditions was not significant (ANOVA: $F(2,123) = 0.71$, $p = 0.496$).

Comparisons were made for each REE leached into solution in the biological experiments compared to the non-biological control for *B. subtilis* and *C. metallidurans* and for each separate gravity condition (*t*-test). In *B. subtilis*, for simulated Mars and Earth gravity, concentrations of bioleached REEs in solution were significantly lower compared to the non-biological control (Supplementary Table 4) at the $p < 0.05$ level, except for Eu, Gd, Tb, Ho and Lu, which were significantly lower at the $p < 0.1$ level, and not significant for Ce in the simulated Earth gravity condition ($p$ value = 0.378). In *C. metallidurans*, Tm, Yb and Lu were statistically lower at the $p < 0.1$ level in simulated Mars gravity (Supplementary Table 4).

Comparisons were also made for each REE leached into solution in the biological experiments between gravity conditions for *B. subtilis* and *C. metallidurans* (*t*-test). In *B. subtilis* cultures, six elements (Dy, Ho, Er, Tm, Yb, Lu) showed a difference at the $p < 0.05$ level between microgravity and simulated Mars gravity and one element (Ce) at the $p < 0.05$ level between simulated Mars and Earth gravity. For *C. metallidurans* cultures, Ce was the only element that showed a significant difference at the $p < 0.01$ level between microgravity and simulated Mars gravity. For both *B. subtilis* and *C. metallidurans*, concentrations of elements leached as a percentage of the total available in the basalt ranged from $3.22 \times 10^{-2}$ to $4.14 \times 10^{-3}$ % (Supplementary Table 5).

To test whether the REEs were absorbed onto the cell membrane or within the microbial cell, ICP-MS analyses of the cell pellets were performed (Supplementary Table 6). The

concentrations of REEs in these samples generally accounted for less than 5% of the total REEs in the bulk solution in the biological experiments, with a few exceptions. Notably, Eu was above 5% in all conditions apart from *S. desiccabilis* in microgravity and Mars gravity. ANOVA was used to ascertain whether the biological enhancement of REEs leached into solution exhibited by *S. desiccabilis* was also reflected in the quantity of REEs bound to cells compared to the two other organisms. In microgravity, there was a significant difference between the organisms (ANOVA: $F_{(2,125)} = 3.98$, $p = 0.021$), but post-hoc Tukey showed that only the *S. desiccabilis* and *B. subtilis* pairwise comparison was significant ($p = 0.016$). There was no significant difference between organisms in Mars gravity (ANOVA: $F_{(2,125)} = 0.466$, $p = 0.629$). In Earth gravity, there was a significant difference (ANOVA: $F_{(2,125)} = 36.94$, $p < 0.001$) with post-hoc Tukey showing $p < 0.001$ for all pairwise comparisons apart from *S. desiccabilis* and *C. metallidurans* ($p = 0.132$). In almost all cases the percentage of total REEs associated with the *S. desiccabilis* cell pellets were lower than the other two organisms (Supplementary Table 6). Thus, there was no evidence for a systematically higher fraction of REEs in the *S. desiccabilis* cell pellets. The concentrations of REEs in the supernatant produced from washing of the cell pellet was below the detection limit.

Comparison of the REEs leached into solution between the different gravity regimens of the non-biological control samples on the ISS (Figs. 2, 3a, Supplementary Fig. 1 and Supplementary Table 2) showed that the gravity condition was not significant (ANOVA: $F_{(2,109)} = 2.91$, $p = 0.059$). Student's *t* test investigations of individual elements in each gravity condition (Supplementary Table 4) showed that Pr, Nd, Sm, Eu, Gd, Tb, Dy were significantly different (at the $p < 0.1$ level) between simulated Mars and Earth gravity control samples. The pure 50% R2A medium and NOTOXhisto fixative contributed low concentrations of REEs (<0.1 ng to the total solution concentration).

**S. desiccabilis caused preferential leaching of heavy REEs.** The percentage difference in bioleaching of REEs was calculated for each microorganism relative to the leaching in the non-biological controls in the same gravity condition, for space and ground experiments (Fig. 3 and Supplementary Table 3).

*S. desiccabilis* caused preferential leaching of heavy (Gd up to Lu) over light (La up to Eu) REEs. On the ISS, the highest enhancement was a $429.2 \pm 92.0\%$ increase in Er leaching in simulated Earth gravity, compared to the non-biological control. On the ground, Yb showed the highest enhancement of $767.4 \pm 482.4\%$ increase in bioleaching over the non-biological control. The larger differences between the non-biological and the biological leaching of heavy REEs compared to light REEs is reflected in generally lower *p* values (Student's *t*-tests) for heavy REEs compared to light REEs (Supplementary Table 4).

**Performance of biomining in space and true Earth gravity.** In parallel with the ISS experiment, ground experiments (true Earth gravity control) were conducted. Results from the ground control experiments are reported in Figs. 2, 3b, Supplementary Fig. 1, Supplementary Tables 2 and 3. For *S. desiccabilis*, the effect of the microorganism on leaching in the ground control compared with the non-biological control was significant (ANOVA: $F_{(1,68)} = 24.56$, $p < 0.001$). All individual elements showed a statistically significant difference (Student's *t* test) with the non-biological control (Supplementary Table 4) at the $p < 0.05$ level apart from two elements at the $p < 0.1$ level (Nd and Sm) and three elements with no significant difference (La, Ce, Eu). For *B. subtilis*, the effect of the microorganism on leaching was not significant

(ANOVA: $F_{(1,68)} = 0.034$, $p = 0.854$), similarly with *C. metallidurans* (ANOVA: $F_{(1,68)} = 0.705$, $p = 0.404$).

Bioleaching of REEs in simulated Earth gravity on the ISS was compared to bioleaching in the ground experiment (true Earth gravity). *S. desiccabilis* showed a significant difference (ANOVA: $F_{(1,82)} = 8.14$, $p = 0.005$) with simulated Earth gravity on ISS being higher across all REEs. Neither *B. subtilis* (ANOVA: $F_{(1,82)} = 2.42$, $p = 0.124$) or *C. metallidurans* (ANOVA: $F_{(1,82)} = 2.45$, $p = 0.121$) showed a significant difference. Non-biological controls exhibited a significant difference between the simulated Earth gravity on the ISS and ground controls (ANOVA: $F_{(1,68)} = 6.90$, $p = 0.011$) with the concentration of REEs leached into solution in simulated Earth gravity on the ISS being higher across all REEs.

**Biomining occurred under near neutral pH conditions.** The pH status is an important factor in the efficacy of biomining. NOTOXhisto fixative lowers the final pH of the solutions, so that the pH at the end of the experiment is not representative of the pH during growth. In all experimental solutions, the final pH ranged between $4.16 \pm 0.20$ and $6.12 \pm 0.01$ (Supplementary Table 7).

As it was not possible to measure the pH during the experiment on the ISS, a ground experiment was conducted to investigate pH changes over the 21 days of growth at 20–22 °C. Results are shown in Supplementary Fig. 2. The pH remained circumneutral for the non-biological samples throughout the experiment with slight differences in the presence of basalt. The presence of bacteria caused the pH to rise during the 21 days compared to the negative controls, regardless of the specific species. At day 21, the pH values for the three cultures in the presence of basalt were: *S. desiccabilis*, $8.41 \pm 0.01$; *B. subtilis*, $8.63 \pm 0.01$; *C. metallidurans*, $8.66 \pm 0.01$, and the non-biological control $7.35 \pm 0.036$ (mean ± sd). The presence of the basalt slide caused slight pH differences within the biological samples during the first week of growth. After one week, the pH remained constant until the end of the experiment for all the microorganisms. After 21 days of growth, the pH values with and without the presence of the rock are similar for each microorganism, suggesting that the influence of the rock material on the pH values stabilised over time (Supplementary Fig. 2). There was a large drop in pH after the addition of the fixative (*S. desiccabilis*, $3.58 \pm 0.07$; *B. subtilis*, $3.89 \pm 0.10$; *C. metallidurans*, $3.76 \pm 0.08$, and the non-biological control $3.08 \pm 0.03$, mean ± sd). The post-fixative pH values are different depending on the organism, but independent of the presence of the basalt. After one week of cold storage, the presence of the basalt slide caused an increase in pH for all biotic and non-biological samples, indicating that the pH measured in the flight and ground control samples was influenced by both the presence of the basalt slide and the fixative.

## Discussion

This study investigated the use of microorganisms to extract a group of economically important elements (fourteen REEs) from basalt rock, a material found on the Moon and Mars[36–38], under simulated Mars and Earth gravity on the International Space Station (ISS). Microgravity was investigated as the lowest gravity level possible to explore the effects of a lack of sedimentation on bioleaching, to understand the role of gravity in influencing microbe–mineral interactions in general, and to gain insights into industrial biomining on asteroids and other very low gravity planetary objects. A true Earth gravity ground control experiment was also performed.

The presence of the bacterium *S. desiccabilis* was found to enhance mean concentrations of leached REEs in all gravity

conditions investigated and these enhancements were significant in simulated Mars and Earth gravity on ISS compared to the non-biological controls. Although the *S. desiccabilis* microgravity samples reached higher mean concentrations than the microgravity non-biological controls for all REEs, the difference was not statistically significant. The statistical result is interpreted to be caused by the greater standard deviations in the leached concentrations of elements in the microgravity biological experiment and non-biological controls and the loss of one of the microgravity control samples owing to contamination, rather than an effect of microgravity on biological leaching.

The lack of a significant difference in the final concentrations of REEs leached by *S. desiccabilis* when the different gravity conditions were compared is surprising since microgravity has been reported to influence microbial processes[39,40]. However, the results are consistent with our observation that final cell concentrations did not differ between the different gravity conditions in the three microorganisms[31]. One reason for the lack of statistically significant differences in final concentrations of REEs between gravity conditions might be that the bacterial cultures had sufficient nutrients to reach their maximum cell concentration[31], regardless of the different sedimentation rates in each gravity, thus achieving similar leaching concentrations. Hence, the experiments showed that, with the appropriate nutrients, biomining is in principle achievable under a wide range of gravity conditions.

The mechanism for the REE bioleaching in *Sphingomonas desiccabilis* is unknown. It was not caused by bulk acidification of the growth medium, since the ground experiments showed that the medium had a slightly basic pH profile during the experiment. The microorganism is a prolific producer of extracellular polysaccharide (EPS) and these compounds are known to enhance bioleaching in other organisms by complexing ions in EPS moieties such as uronic acid[41,42]. A greater biological enhancement in the leaching of heavy compared to light REEs was observed, a pattern consistent with observations by Takahashi et al.[43] in laboratory cell cultures and natural microbial biofilms. The authors suggested that phosphate moieties on the cell or EPS might preferentially bind heavy REEs, a distinct property of these biologically produced materials. We also note that the authors suggested that heavy REE enrichments could potentially be used as a biosignature for the activities of life. Beyond applications to biomining, our experiments showed the preferential enhancement of heavy REEs in the liquid phase including in simulated Martian gravity, indicating the production of a potential biosignature under altered gravity, with implications for example for additional methods to test the hypothesis of life on Mars.

Enhanced REEs associated with pelleted *S. desiccabilis* cells compared to the other two species was not observed. The reduced pH caused during fixation and sample preparation may have unbound any REEs attached to cell surfaces in all three species. Alternatively, the majority of the REEs may have bound to the extracellular EPS or have been released directly into solution. We have observed *S. desiccabilis* by confocal microscopy to form biofilms on the surfaces and at the edges of cavities on the basalt more pervasively than *B. subtilis* and *C. metallidurans* under these growth conditions, which could have enhanced cell-mineral interactions and thus leaching of REEs into solution. The analysis of REEs within biofilms did not form part of this study since we wished to separately examine the biofilms non-destructively.

Unavoidable in this experiment was the potential for continued leaching after fixation and during storage, when the pH was reduced in the chamber. However, during storage, the temperature was kept at 2.1 °C on the ISS and below 7.1 °C during sample return to reduce leaching activity[44]. Furthermore, a similar reduction of the pH occurred in the non-biological control samples.

In contrast to *S. desiccabilis*, *B. subtilis* demonstrated less mean leaching in the biological experiments than the non-biological controls in all three gravity conditions. This cannot be attributed to cells attached to the rock retarding ion release since the microorganisms did not form substantial biofilms on the surface of the rock and the final cell biomass was lower than in the case of *S. desiccabilis*[31]. As the pH was likely to be similar to the other organisms during the course of the experiment as shown by our ground-based post-flight pH experiment, differences in pH during the experimental phase cannot explain the results. An alternative explanation could be a chemical effect of cell exudates, such as ligands that retarded leaching or the solubility of REEs. However, despite its previously demonstrated bioleaching activity[45,46], and cell wall absorption of REEs[47], Kucuker et al.[48] showed that *B. subtilis* was not able to extract tantalum, a transition metal considered similar to a REE, from capacitors.

*C. metallidurans* did not enhance leaching of REEs. In a 3-month preparatory phase for the BioRock experiments, the leaching of elements from crushed basalt by this organism on the Russian FOTON-M4 capsule was investigated[49]. In this experiment, *C. metallidurans* enhanced copper ion release, but other rock elements did not show significantly enhanced leaching. Although the microorganism was suspended in mineral water, the results are consistent with those reported here.

In none of the experiments was a cerium anomaly[50] observed. Unlike other REEs that are all trivalent, cerium can be oxidised to the less soluble $Ce^{4+}$ state, which can cause differences in precipitation and concentration compared to other REEs. The experiments were performed under oxic conditions. However, once the cerium was leached from the rock, its oxidation state would not necessarily have changed its presence in the bulk fluid, potentially explaining the lack of an anomaly.

Comparing the Earth gravity simulation on the ISS with the ground-based experiments (true $1 \times g$ control), no significant difference was observed between biological experiments with *B. subtilis* and *C. metallidurans*, but there was a significant difference between the *S. desiccabilis* biological experiments and between the non-biological controls, with ground-based leaching significantly less in some REEs compared to the Earth gravity simulation on the ISS. Simulated gravity in space is not exactly the same as $1 \times g$ on Earth as shear forces induced by centrifugation in space can create different physical conditions. Furthermore, because of the small radius of the centrifuge rotor in KUBIK, gravity forces vary across the culture chamber. We also note that the ground experiment had a 0.46 °C higher temperature offset than the KUBIKs on the ISS during the main experimental phase. The experiment on the ISS involves the launch and download to Earth of the samples, which could influence them in ways that cannot be easily predicted. Nevertheless, the general trends observed in Earth gravity experiments with respect to biologically enhanced leaching for the three organisms were conserved in space.

Our experiment has several differences with any proposed large-scale biomining activity. The basalt rock was not crushed in order to investigate biofilm formation on a flat, contiguous but porous rock surface, another main goal of the BioRock experiment. This may have influenced the total percentage of REEs extracted from the rock, which was generally less than $5 \times 10^{-2}$ %. These leaching rates would likely be higher with crushed rocks, which on Earth have been shown to result in leaching efficiencies of REEs of $8.0 \times 10^{-3}$% to several tens of percent under optimised conditions[51,52]. Furthermore, we did not stir our reactors as we wanted to investigate the effects of microgravity and Mars gravity on cell growth in the absence of artificial

mixing. Understanding which parameters would require adjustments to enhance the process as well as upscaling of the reactor would be the next step. Our experiment demonstrates that the leaching capacities of the three different microorganisms on the Earth[53,54] were similar in space. Thus, Earth-based ground experiments provide reliable insights into the biomining capacities of specific organisms in space. Yet, our experiments also confirm that it is important to be careful in the selection of microorganisms for space biomining operations.

Basaltic material was investigated because it is common on the Moon and Mars[36–38]. Our experiment suggests that other materials could return even higher yields. For example, lunar KREEP rocks have unusually high concentrations of REEs[55,56]. We did not test lunar gravity ($0.16 \times g$) directly, but it lies between microgravity and Mars gravity. Our results therefore likely reflect the potential efficacy of biomining operations under lunar gravity. We suggest the construction of REE biomining facilities in the Oceanus Procellarum and Mare Imbrium regions of the Moon, where KREEP rocks are abundant. The principle we demonstrate could be applied to other materials of economic importance for In-Situ Resource Utilisation (ISRU). For example, meteoritic material has been shown to be compatible with microbial growth[26,57–60] and thus our microgravity experiments show the potential for biomining in low gravity asteroid environments.

In conclusion, our results demonstrate the biological mining of economically important elements in space, specifically REEs and in different extraterrestrial gravity environments. The experiments also demonstrate the novel REE bioleaching ability for the mesophilic, biofilm-forming, and desiccation-resistant bacterium *S. desiccabilis*, which could be used in biomining applications. From a technical point of view, our experiment also demonstrated the principles of a miniature space biomining reactor. The experiment thus shows the efficacy of microbe–mineral interactions for advancing the establishment of a self-sustaining permanent human presence beyond the Earth and the technical means to do that.

## Methods

**BioRock experiment.** BioRock was an experiment proposed to European Space Agency (ESA) in response to the International Life Science Research Announcement in 2009 (ILSRA-2009). The project was selected as a candidate for flight in 2010 and subsequent bioreactor hardware design has been described[61]. The experiment began on the International Space Station on July 30, 2019 and ended on August 20, 2019.

**Microorganisms and growth media.** Three bacterial species were used to investigate bioleaching. Criteria were: (1) they could tolerate desiccation required for experiment preparation, (2) they could grow on solid surfaces and/or form biofilms, (3) they were able to interact with rock surfaces and/or bioleach, and (4) they all could be grown in an identical medium at the same experimental conditions to allow for comparisons between organisms.

The microorganisms used were:

*Sphingomonas desiccabilis* CP1D (DSM 16792; Type strain), a Gram-negative, non-motile, desiccation resistant, non-spore-forming bacterium, which was isolated from soil crusts in the Colorado plateau[62].

*Bacillus subtilis* NCIB 3610 (DSM 10; Type strain), a Gram-positive, motile, spore- and biofilm-forming bacterium naturally found in a range of environments, including rivers[63]. The organism has been used in several space experiments[28,33].

*Cupriavidus metallidurans* CH34 (DSM 2839; Type strain), a Gram-negative, motile, non-spore forming bacterium. Strains of this species have been isolated from metal-contaminated and rock environments[64–69]. The organism has been previously used in space experiments[70].

The medium used for the BioRock experiment was R2A[71] at 50% concentration as it supported growth of all three microorganisms[61], allowing for comparisons. The composition was (g L$^{-1}$): yeast extract, 0.25; peptone, 0.25; casamino acids, 0.25; glucose, 0.25; soluble starch, 0.25; Na-pyruvate, 0.15; K$_2$HPO$_4$, 0.15; MgSO$_4$.7H$_2$O, 0.025 at pH 7.2.

NOTOXhisto (Scientific Device Laboratory, IL, USA), a fixative compatible with safety requirements on the International Space Station (ISS), was used to halt bacterial metabolism at the end of the experiment.

**Bioleaching substrate.** Basalt was used for bioleaching, whose REE composition, as determined by ICP-MS (inductively coupled plasma mass spectrometry) and bulk composition, as determined by X-ray Fluorescence (XRF), is shown in Table 1 and Supplementary Table 1, respectively. The material was an olivine basalt rock collected near Gufunes, Reykjavik in Iceland (64°08′22.18″N, 21°47′21.27″W) chosen because it has a chemical composition similar to that of basalts found on the Moon and Mars[36–38]. The rock was cut into slides of 1.5 cm × 1.6 cm and 3 mm thick. The mass of 15 of these slides was 1.871 ± 0.062 g (mean ± standard deviation). The rock was not crushed, as might be carried out for large-scale bioleaching, because the BioRock project was also concerned with quantifying the formation of microbial biofilms over a contiguous mineral surface.

**Sample preparation for flight.** The basalt rock slides were sterilised by dry-heat sterilisation in a hot air oven (Carbolite Type 301, UK) for 4 h at 250 °C. This treatment did not change the mineralogy of the rocks as determined by X-Ray Diffraction (XRD).

Single strain cultures of each organism were desiccated on the slides as follows: *S. desiccabilis*. An overnight culture of the strain was grown in R2A 100% at 20–22 °C until reaching stationary phase (OD600 = 0.88 ± 0.09; approximately 10$^9$ colony forming units per mL). Then, 1 mL of the culture was inoculated on each basalt slide and the samples were air-dried at room temperature (≈20–25 °C) with a sterile procedure within a laminar flow-hood.

*B. subtilis*. Spores were produced as described previously[72]. For each basalt slide, 10 μL of a ≈ 1 × 10$^8$ spores/mL solution were used as inoculum, i.e. 1 × 10$^6$ spores per slide, and air-dried at room temperature (≈20–25 °C) within a laminar flow-hood.

*C. metallidurans*. Samples were prepared using a freeze-dry protocol (Belgian Co-ordinated Collection of Microorganisms, BCCM). Cells were cultured on solid Tryptone Soya Agar (TSA, Oxoid CM0131, BCCM) medium. When grown confluently, cells were harvested with a cotton swab and suspended in cryoprotectant, consisting of sterile horse serum supplemented with 7.5% trehalose and broth medium no. 2 (625 mg L$^{-1}$; Oxoid CM0131, BCCM). Thirty millilitres of bacterial suspension were transferred to a 90 mm petri dish and basalt slides were submerged in the bacterial suspension and gently shaken overnight. Basalt slides, each containing approximately 10$^9$ colony forming units per mL, were then transferred to a 6-well plate (1 slide per well) and covered with a gas permeable seal and inserted on a pre-cooled shelf of −50 °C, followed by a freezing phase for 90 min at a shelf temperature of −50 °C. Primary drying was performed with a shelf temperature of −18 °C and chamber pressure of 400 mTorr. A secondary drying was performed with a shelf temperature of 20 °C and a chamber pressure below 10 mTorr. After freeze-drying, the 6-well plate was covered with a lid and wrapped in parafilm.

Negative controls were sterile basalt slides without cell inoculation.

After preparation, all samples were stored at room temperature (20–25 °C) until integration in the culture chambers in the bioreactor.

**Flight experimental setup.** The hardware design, assembly and filling procedure were described previously[61]. Each Experiment Unit (EU) of the BioRock apparatus was designed to accommodate two independent basalt slides in two independent sample chambers (Fig. 1). Each EU contained culture medium and fixative reservoirs (Fig. 1a). To allow oxygen diffusion without contaminating the cultures, each chamber was equipped with a deformable, gas permeable, silicone membrane (Fig. 1b, c)[61]. After integration of the basalt slides, the medium and fixative reservoirs were filled with 5 mL of medium and 1 mL of fixative for each sample, respectively. The culture chambers and surrounding ducts were purged with ultrapure sterile N$_2$ gas.

All the samples were integrated under strict aseptic procedures into the EUs. There were 36 samples in 18 EUs for the flight experiment, and 12 samples in 6 EUs for the ground experiment. The EUs were integrated into a secondary container that provided the required two-level containment of the fixative (Fig. 1a). The EU within the container is referred to as the Experiment Container (EC).

After integration, 18 flight ECs were stored at room temperature (≈20–25 °C) for 2 days. The ECs were launched to the ISS on board of a Space X Dragon capsule, Falcon-9 rocket during CRS-18 (Commercial Resupply Services) mission on July 25, 2019 from the NASA Kennedy Space Center, Cape Canaveral, Florida. On arrival at the ISS, ECs were stored on-board at 2.1 °C.

On the day of the start of the experiment (July 30, 2019), the ECs were installed by astronaut Luca Parmitano into two KUBIK facility incubators, pre-conditioned to a temperature of 20 °C (Fig. 1d). Medium injection was performed robotically, triggered by internal clocks built within the ECs, powered by electricity provided by the KUBIK incubator. Thereafter the astronaut removed the ECs and took photographs of all culture chambers to obtain evidence of the medium supply and to allow comparison with the same chamber after the experimental growth period. After image acquisition, the ECs were plugged back into the KUBIKs. Two KUBIK incubators were used for BioRock, running in parallel: One was set to simulate Earth gravity ($1 \times g = 9.81$ m/s$^2$) at the surface of the basalt slide where bioleaching is occurring, while the second was set to simulate Mars gravity ($0.4 \times g = 3.71$ m/s$^2$; Mars gravity is strictly $0.38 \times g$, but finer $g$ resolution is not possible to set in the KUBIK) at the surface of the basalt slide. Gravity levels were measured every 10 min during the active experimental phase using an accelerometer (ADXL313,

Analog Devices) mounted on a printed circuit board fixed to the bottom of the EC. The distance between the top face of the basalt slide and the plane of the top face of the PCB was 10.3 mm. A correction factor was applied to account for the longer rotation radius at the basalt slide. These accelerometer (gravity) values are shown in Supplementary Table 8. The microgravity-exposed ECs were split equally between both KUBIKs and inserted in the static slots available in the facility. The experiment was run for 21 days.

To stop the cultures from growing, the fixative was automatically injected into the culture chambers on August 20, 2019. The samples were removed from the KUBIK incubators and images of the culture chambers were taken. Afterwards, the ECs were stored in refrigeration as described below.

**Temperature during space experiment**. The temperature of the ECs from pre-flight until post-flight was measured using temperature loggers (Signatrol SL52T sensors, Signatrol, UK) on the rear of four of the ECs. These data showed that temperatures did not exceed 7.1 °C from pre-flight handover until storage after arrival at the ISS. During on-board storage, both before and after the 21-day period of culturing, temperatures were constant at 2.1 °C. During culturing, the loggers recorded a temperature of 20.16 °C in both KUBIKs. The ECs were downloaded from the ISS, packed in a 'double coldbag' provided by NASA. Splashdown occurred in the Pacific Ocean on August 27 and handover of the ECs to the investigators occurred on August 29 at Long Beach Airport, LA, USA. Between removal from storage on August 26 on ISS and handover to the science team on August 29, the temperature loggers recorded a temperature of 6.6 °C, rising transiently to 7.1 °C. The ECs were stored in a refrigerated insulated box and transferred to the NASA Ames Research Centre for sample removal on August 30.

**Ground experiment**. Parallel to the experiment occurring on the ISS, a $1 \times g$ ground experiment (true Earth gravity) was run to compare with the ISS simulated Earth gravity samples. Six ECs for the ground experiment were shipped from the NASA Kennedy Space Center to the NASA Ames Research Centre under cooled (4 °C) conditions. The six ECs were attached to a power system (KUBIK Interface simulation station, KISS) with leads running from the system into a 20 °C laboratory incubator (Percival E30BHO incubator). The ground reference experiment commenced 2 days after the start of the space experiment and the procedure for the space experiment was replicated: medium injection, first image acquisition, 21-day experiment, fixation, second image acquisition, and cold storage at 4 °C. The temperatures of the ECs measured by temperature logger (see above) on two of the ECs were 3.58 and 4.54 °C during shipment to NASA Ames. During the 21 days main experimental phase, the loggers recorded a temperature of 20.62 °C. During post-experiment storage, the temperature was 3.06 °C.

**Sample recovery**. Liquid and basalt slide removal from the ECs was performed at the NASA Ames Research Center. From the total of 6 mL of total bulk fluid per EC, an aliquot of 3 mL was taken and 65% nitric acid was added to a final concentration 4% to fix ions and minimise attachment and loss to container walls. These samples were cold stored at 4 °C until further analysis.

Fixative injection was successful for all the space ECs. However, fixative injection failed in four of the ground experiment chambers: one *B. subtilis* chamber, two *C. metallidurans* chamber and one non-biological control sample. In these cases, 1 mL of NOTOXhisto was added to the liquid samples before the abovementioned procedures.

In all ECs, two culture chambers were observed to have contamination: an ISS non-biological control chamber in microgravity, juxtaposed to a *B. subtilis* chamber, was contaminated with cells that were morphologically identical to *B. subtilis* chamber. In the ground control samples, a non-biological control chamber, juxtaposed to a *B. subtilis* chamber, had a cellular contaminant at low concentration that formed a white pellet on centrifugation that was morphologically dissimilar to *B. subtilis*. NOTOXhisto fixation prevented successful DNA extraction and identification in both cases. These data points were removed from the calculations.

All samples were shipped back to the University of Edinburgh in cold storage by Altech Space (Torino, Italy).

**ICP-MS analysis of samples**. Upon return to Edinburgh, UK, the 3 mL of acid-fixed sample was prepared in the following way: each sample was sequentially (in three batches) spun down in a 1.5 mL tube at 10,000 × $g$ (IEC MicroCL 17 centrifuge, Thermo Scientific) for ten minutes to pellet cells and cell debris. The supernatant was collected into a 15 mL tube and analysed by ICP-MS (inductively coupled plasma mass spectrometry). Acquired liquids were used to determine the bulk fluid REE concentrations. Cell debris pellets were washed two times in ddH$_2$O and this discarded liquid was pooled. Nitric acid was added to the pooled fluid to a final concentration of 4%, and the samples were analysed by ICP-MS. This liquid was used to determine the REE concentrations that were washed off the cell matter. The pellet was transferred to an acid-washed glass serum bottle pre-baked at 450 °C in an oven (Carbolite Type 301, UK) for 4 h to remove organic molecules. The vial with the pellet was heated at 450 °C for a further 4 h to volatilise carbon and leave residual ions. After cooling, 1.5 mL of ddH$_2$O were added with nitric acid to a final

concentration of 4% and samples were analysed by ICP-MS. This liquid was used to determine the REE concentrations associated with the cell material.

ICP-MS analysis was carried out as described below on the R2A 50%, NOTOXhisto and ddH$_2$O. It was not possible to examine the separated cryoprotectant for *C. metallidurans*. However, as a significance enhancement in the biological sample compared to the controls was not observed, we infer that the protectant did not add additional REEs.

All samples were analysed by ICP-MS using an Agilent 7500ce (with octopole reaction system), employing an RF (radio-frequency) forward power of 1540 W and reflected power of 1 W, with argon gas flows of 0.81 L/min and 0.20 L/min for carrier and makeup flows, respectively. Sample solutions were taken up into the micro mist nebuliser by peristaltic pump at a rate of approximately 1.2 mL/min. Skimmer and sample cones were made of nickel.

The instrument was operated in spectrum multi-tune acquisition mode and three replicate runs per sample were employed. The isotopes: [139]La, [140]Ce, [141]Pr, [146]Nd, [147]Sm, [153]Eu, [157]Gd, [159]Tb, [163]Dy, [165]Ho, [166]Er, [169]Tm, [172]Yb, [175]Lu were analysed in 'no gas' mode with each mass analysed in fully quantification mode and three points per unit mass. The REE Pm (promethium) was not measured as the element is radioactive and no standard was available.

To calibrate the instrument, multi-element calibration standards containing each element were prepared using an REE multi-element standard (Multi-Element Calibration Standard-1, Agilent Technologies, USA) plus a Uranium single-element 1000 mg L$^{-1}$ standard (SPE Science, Canada) diluted with 2% v/v HNO$_3$ (Aristar grade, VWR International, United Kingdom). A NIST standard reference material, SRM1640a, was employed as a reference standard for some of the elements. The limits of detection for the REEs were split broadly into two groups: La, Ce, Pr, Tb, Ho, Tm, Lu: 0.0025-0.005 ppb. Nd, Sm, Eu, Gd, Dy, Er, Yb: 0.001-0.005 ppb.

Raw ICP-MS data (determined in µg/L) was converted to obtain the absolute quantity of a given element in the culture chamber, taking into account dilution factors applied during ICP-MS analysis.

To determine REE concentrations in the basalt slide, between 25 and 50 mg of homogenised sample was added to Savillex Teflon vessels. Rock standards (basalt standards BIR-1, BE-N, BCR-2, BHVO-1) were prepared in the same way. Two blanks were included (i.e., sample without basalt). Three millilitres of double distilled HNO$_3$, 2 mL HCl and 0.5 mL HF was added to each of the vessels. HF was added after the other acids to prevent disassociation, formation and precipitation of aluminium fluorides. Samples were placed on a hot plate for digestion overnight (temperature of 100–120 °C) and checked for complete digestion. Samples were evaporated on the hot plate. Five millilitres of 1 M HNO$_3$ was added to each vessel. Lids were added and the samples returned to the hot plate for a second digestion step. Samples were further diluted with 2–5% HNO$_3$ for ICP-MS analysis.

Analysis was carried out on a high resolution, sector field, ICP-MS (Nu AttoM). The ICP-MS measurements for REEs were performed in low resolution (300), in Deflector jump mode with a dwell time of 1 ms and 3 cycles of 500 sweeps. Data were reported in micrograms of REE per gram of basalt.

**pH of flight experiment and ground pH experiment**. A small aliquot (≈0.3 mL) of liquid from the chambers was used to measure the pH of the solutions at the end of the experiment after fixative addition. The pH was measured using a calibrated Mettler Toledo Semi-Micro-L pH metre. Final values for cell growth in the experiment are reported previously[31] and the values are shown in Supplementary Table 9.

During the space experiment, only final pH values were obtained. Thus, an experiment was carried out on the ground to investigate the pH changes that might have occurred during the course of the experiment (limited to a $1 \times g$ condition) and the influence of the basalt rock in any observed pH changes.

Sterile basalt slides, as used in the flight experiment, were prepared in 5 mL of 50% R2A in six-well plates (Corning, UK) and the wells were inoculated with one of the three microorganisms used in this study. Control experiments were conducted without organisms, using fresh 50% R2A only, with or without the basalt slide. The experiment was performed at 20 °C for 21 days. After this period, 1 mL of NOTOXhisto was added to each well, and stored at 4 °C for a further week. During the course of the experiment, pH values were measured at fixed intervals (day 0, 1, 4, 7, 14, 21). On the 21st day of the experiment, pH was measured twice, before and after the fixative addition.

**Statistical analysis and software**. Analysis of the leaching data was performed at several levels of granularity using SPSS Statistics (IBM, version 26). Two and one-way ANOVAs were used to assess significant differences between gravity conditions, organisms, ground and space samples, and between controls, in combinations described in the results. In these analyses, data across all REEs was used (the tests did not discriminate REEs). Data were log$_{10}$ transformed and tests for normality of data and equal variances (Levene's tests) were carried out. Tukey tests were performed where appropriate to examine pairwise comparisons.

To investigate differences between gravity conditions and organisms or controls for specific REEs, a two-sample independent two-tailed Student's $t$ test was used between pairs of conditions for specific REEs, accepting that the small sample sizes make these tests less reliable than the aggregate ANOVA analyses.

RStudio 1.2.5033 was used to analyse and visualise the ground pH experiment. Microsoft Excel (2016) was used to collect data and Microsoft Word (2016) was used to prepare the manuscript and associated text files.

## Data availability

The presented data in the paper are available on the Edinburgh Datashare repository at https://doi.org/10.7488/ds/2908 and from the corresponding author.

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

## Acknowledgements

C.S.C., R.S. and the preparation of the experiment and post-flight analysis were funded by UK Science and Technology Facilities Council under grant ST/R000875/1. AW was supported by a Principal's Career Development PhD Scholarship. R.M., F.M.F. and P.R. were supported by the DLR grant "DLR-FuE-Projekt ISS LIFE, Programm RF-FuW, Teilprogramm 475". F.M.F. was also supported by the Helmholtz Space Life Sciences Research School at DLR. R.V.H. and N.L. received financial support for this study from Belspo and ESA through the PRODEX EGEM/Biorock project contract (PEA 4000011082). We thank Laetitia Pichevin for ICP-MS analysis of the basalt substrate. We thank the European Space Agency (ESA) for offering the flight opportunity. A special thanks to the dedicated ESA/ESTEC teams, Kayser Italia s.r.l., and the USOC BIOTESC for the development, integration and operation effort. We are thankful to the UK Space Agency (UKSA) for the national support to the project, NASA Kennedy for their support in the experiment integration prior to the SpaceX Falcon 9 CSR-18 rocket launch, particularly Kamber Scott and Anne Currin, and NASA Ames for hosting the ground control experiment. We thank SpaceX and Elon Musk for launching our mining experiment into space.

## Author contributions

C.S.C. conceived the BioRock experiment in the framework of the ESA topical team Geomicrobiology for Space Settlement and Exploration (GESSE). C.S.C., R.S. and K.F. designed the experiments for this paper. N.N. and C.M.L. carried out ground experiments and studies in preparation for flight. C.S.C., R.S. and A.C.W. integrated the hardware for spaceflight and ground controls. C.S.C., R.S. and L.J.E. produced the experimental data. C.S.C. and R.S. performed the data analyses. R.M., P.R., F.F. and R.V.H., N.L., I.C. provided *B. subtilis* and *C. metallidurans* samples, respectively. L.P. performed the procedures on-board the ISS. R.D., J.H., J.K., A.K., N.C. and L.Z. supervised the technical organisation and implementation of the experiment at ESA. J.D.W., M.H. and B.R. supervised the flight procedures. A.M., S.P., F.C., G.L., M.B. and V.Z. designed and fabricated the hardware. R.C.E. and J.W. hosted the ground control experiment. C.S.C. wrote the manuscript. All authors discussed the results and commented on the manuscript. C.S.C. and R.S. contributed equally to the work.

## Competing interests

Authors V.Z., M.B., A.M., S.S.P., F.C. and G.L. were employed by the company Kayser Italia S.r.l. The remaining authors declare no competing interests.
