## [Peer Review File · Nature Communications]

REVIEWER COMMENTS

Reviewer #1 (Remarks to the Author):

This is a very interesting report of an investigation of the potential for biomining to be employed in an extraterrestrial environment. Here, the authors employ both a flight-based and accompanied by a ground-based investigation of several representative cultures to extract rare Earth elements (REEs) under microgravity and partial gravity conditions. Given that a number of REEs are associated with meteorites and other extraterrestrial environments, a proof-of-concept study is certainly relevant. Here, the authors employed three test organisms, that were chosen for their abilities to withstand desiccation, form biofilms, grow on a common test medium, and interact with the rock surfaces. The authors provide a very detailed description of the various experimental conditions encountered during the flight, which is highly relevant to understanding the challenges involved. Overall, the data does support the proof of concept. I have some minor considerations that I would like the authors to consider:

1. In lines 92-94, the authors claim that prokaryotes are unable to directly sense gravity. There are a number of published experiments that would counter this claim (e.g. work by CA Nickerson, A Matin, and others).
2. In line 485, the authors mention that dry heat sterilization did not alter the mineralogy of the basaltic rocks. While likely a very minor concern, what data supports this claim?
3. The authors are to be commended for the very thorough characterization of the experimental issues (contamination, gravity variance due to the ISS-based centrifuge rotor size, temperature fluctuations etc.) that occurred during the flight experimentation.
4. While the work presented supports the proof-of-concept of microbial mineral recovery of rare Earth elements (REE) in extraterrestrial environments, the experimentation as presented represents several organisms grown as monocultures. Realistically a biomining setup would involve some type of polymicrobial association. In this context, some of the contaminated samples originally not included in this analysis, might provide an anecdotal insight into this possibility, and as a suggestion the authors may consider including some of this data in the supplemental information. Obviously, a full investigation would entail an analysis of the organisms that are present (i.e. metagenomic analysis) which would be outside the full scope of the reported work. This comment is intended as a suggestion.

Robert JC McLean

Reviewer #2 (Remarks to the Author):

Comments for manuscript NCOMMS-20-26738

General comments:

The manuscript explored bioleaching of REE under microgravity and Earth gravity conditions at International Space Station and on Earth. The results are novel and of interest to both space and biomining communities.

In a number of places the statements of the authors need clarification on the context to make it more clear to the reader what they are referring to.

The solubilised REE concentrations are very low (about 1000 times lower than reported in some previous REE bioleaching studies). The authors should compare the leached concentrations to the concentrations reported to be bioleaching in previous studies from various REE minerals. The authors should also comment on whether such low concentrations as reported in this study would really be economical to recover using the proposed bioleaching approach.

Throughout the manuscript, please do not use active expressions such as "We investigated", "we

found", "our experiment" but rather use passive expressions "was investigated", "was found", "the experiment" instead

References, please provide all authors for each of the references in the reference list, instead of saying et al.

Specific comments

Lines 34 and 70 and 103 and throughout the manuscript, please use small initial letters for "rare earth elements"

Line 61, "accelerates" compared to what? Please specify

Lines 61-63, The sentence is too simplistic. Please note that commercial operations which utilise microbes for biooxidation of refractory gold ores and concentrates as a pre-treatment still typically rely on use of cyanide for gold solubilisation. Hence in the case of refractory gold ores and concentrates, the use of microbes does not typically avoid cyanide use. Moreover, copper is not typically extracted with cyanide. Rather traditional processes have been flotation followed by smelting and chemical leaching. Please revise the text accordingly.

Line 63, which "mechanism" are the authors referring to? Several different mechanisms can be used to bioleach metals, e.g. acidolysis, redoxolysis, complexolysis, or e.g. contact leaching, non-contact leaching and cooperative leaching.

Line 64, please add "in" before "a process called"

Lines 65-68, the statements are somewhat conflicting. If the bioleaching by heterotrophs relies on proton or organic acid production and decrease of pH, how can the heterotrophs be effective in bioleaching at circumneutral pH?

Lines 103 and 108, please replace "Rare Earth Elements (REEs)" with "REEs" as the term has already been previously defined

Line 104, please remove "and biomining" as it is redundant when bioleaching has already been mentioned

Lines 108-109, the statement is too simplistic as based on abstract "Bacillus subtilis" did not enhance leaching as compared to abiotic control although "Sphingomonas desiccabilis" did and no difference was observed for "Cupriavidus metallidurans". In fact the whole paragraph does not mention which of the three species the results refer to. Please amend the paragraph to specify what species they refer to.

Figure 1b, please clarify what the yellow colour in the figure indicates

Table 1, Please replace "concentration" with "content" (in caption and table) when talking about REE content in solids such as basalt, as the word concentration is typically used for solutions. It is also confusing that the authors say concentration (ng in chamber fluid volume of 6 mL). Do the values represent REE "mass" in 6 mL or have the authors calculated a concentration in the unit of ng/mL? Suggest using the latter to avoid confusion and adding the unit also into the table to improve clarity.

Figure 2, the horizontal axis title and values are too small to read. Please increase font size to improve readability

Figure 2, the figure has text "ground experiment", but does not show which data is from "experimental" flight". Please clarify the distinction in the figure

Line 162, the authors should clarify what the percent values refer to, e.g. "relative % difference in mean of biological experiments as compared to controls" as now the meaning is not clear

Figure 3, please use thicker lines for the figure borders (e.g. axis), similar to those used for the legend box.

Supplementary table 2, please clarify in table caption whether the REE quantities are for soluble or total REEs. Please add headings for the first two columns and indicate in table which experiments were done at ISS

Supplementary table 3, Please add headings for the first two columns and indicate in table which experiments were done at ISS

Supplementary table 4, please clarify to what data the differences refer to as it has not been specified in the caption. Also please indicate in the table which experiments were done at ISS

Supplementary table 6, and indicate in table which experiments were done at ISS and clarify how the % was calculated. The current description "expressed as a percentage of total in the pellet and bulk

chamber fluid (%)” is confusing. Do you mean “mass in pellet x100% / (mass in pellet + mass in bulk fluid)”?

Supplementary table 7, please indicate in the table which experiments were done at ISS

Lines 192-193, the statement is confusing. “difference was “significantly less” than what? Do the authors mean that bioleaching resulted in significantly lower REE solubilisation than control leaching both in microgravity and simulated Mars gravity, but under Earth gravity the bioleaching was not significantly different from that of the control? Please revise for clarity.

Lines 208-210, the sentence sounds like the authors would be referring to REE contents in the *C. metallidurans* cells and refer to Supplementary table 4, which does not state for what parameters the p values are shown. The authors should clarify the sentence to indicate that they refer to soluble concentrations instead of REEs in the cells to avoid confusion. Also Supplementary Table should clearly state from what parameters (e.g. soluble REE concentrations) the p values have been derived.

Lines 211-216, the authors should specify whether they talk about soluble REE concentrations or something else as it is not clear from the paragraphs as it currently stands

Lines 217-219, a one sentence paragraph is too short to stand on its own. Suggest combining with the paragraph where the % leaching is given for the third species.

Line 221, please spell out abbreviation “ICP-MS” when first time mentioned

Lines 222-223, the authors state “The concentrations of REE in these samples generally accounted for less than 5 % for all the elements in all the cultures on the ISS,” With “all the elements in all the cultures”, do the authors mean also elements such as carbon, nitrogen, hydrogen, oxygen, phosphorus? Or do the authors mean “% of REEs leached into culture medium”. Please clarify as the sentence is very confusing.

Lines 226-231, please clarify what differences the authors are referring to? % of REEs accumulated in cells? Or something else?

Line 232, what do the authors mean by pellet concentrations? Please clarify as the term is confusing. One could think that the authors are referring to cell numbers, or content of REEs in pellets, but I assume this is not the case. Do the authors mean the % of leached REEs that accumulated in cells?

Lines 234-235, what do the authors mean by “supernatant produced cell pellet preparation”? please clarify

Lines 236-241, it is not clear what parameter the authors are discussing here in terms of differences in gravity conditions. Please clarify if the authors are comparing soluble REE concentrations or something else. Full stop is missing at the end of the paragraph.

Line 250, please specify which REEs are considered as heavy and which as light

Line 262, please replace “compated” with “compared”

Lines 267-269, please combine with previous paragraph, as one sentence paragraph is too short to stand on its own.

Lines 270-282, please clarify if the authors are comparing soluble REE concentrations or something else as it is not clear

Lines 280-282, please combine with previous paragraph, as one sentence paragraph is too short to stand on its own.

Line 218-282, what do the authors mean by “with simulated Earth gravity being higher”? Please clarify if the authors are referring to soluble REE concentrations being higher under simulated Earth gravity than ...

Supplementary Table 7, the authors should clarify in table caption whether the pH was measured after addition of the fixative which changed the pH to avoid misunderstandings.

Lines 293-294, how do the authors explain the increase in pH in the presence of microbes. Previously they mentioned that the bioleaching with heterotrophs relies on production of protons and organic acids

Line 319, please abbreviate genus name as it has been defined before

Line 344, what do the authors mean by “enhancement of the heavier REEs”? please clarify whether

they refer to enhancement of leaching of heavier REEs and as compared to what? it is not clear from the text as it stands

Lines 357-358, please clarify how biofilm formation was observed?

Line 359, please abbreviate genus names as these should have been defined earlier

Line 421, please remove "faithful"

Line 435, please abbreviate genus name

Lines 455-456, please specify whether *Sphingomonas desiccabilis* is motile or not. In the discussion section, please discuss whether the mobility or non-mobility of cells plays a role on the possible impact of gravity on the cells and bioleaching.

Line 474, please add ", respectively" after "Table 1".

Lines 487-490, what was the number of cells used for *S. desiccabilis*?

Line 492, please add "spores" before "per slide"

Lines 494-506, please specify the number of cells per slide for *C. metallidurans*

Line 542, please spell out abbreviation "PCB"

Line 551, please specify temperature for refrigeration

Lines 582-584, were the samples not filtered before nitric acid analysis? How did the authors differentiate between REEs accumulated in cells and REEs that were soluble in bulk medium? Wouldn't acidifying the samples before cell removal result in some REE being resolubilised/desorbed from cells into bulk solution?

Lines 595-596, the sentence does not read well as there is only "In" outside the parenthesis. Please revise for readability

Line 622, please spell out abbreviation "RF"

Line 642, poor English "and took into account", please revise. Also please combine the sentence with previous paragraph as one sentence is too short as a paragraph to stand on its own.

Reviewer #3 (Remarks to the Author):

This manuscript describes an experiment undertaken on the ISS investigating the bioleaching of REEs from basaltic rock under three different gravity conditions as well as a similar experiment on the ground at true Earth gravity. The data presented appears to be a subset of the full data set collected from the experiment (see ref #30).

While interesting, the leaching data on its own is not compelling enough for publication in Nature Communications. A more interesting story could be told by including the microbiological growth and biofilm formation under the same conditions.

The amounts of REEs leached from the basaltic rock are minuscule at 0.00001% albeit significantly different from the abiotic control in the case of *Sphingomonas desiccabilis*.

I query why the three particular microorganisms were chosen for this work. With the exception of a few REEs bioleached by *Sphingomonas desiccabilis*, leaching was not significantly different from the abiotic control even in the ground experiment. There have been many studies in the last 10 years investigating the ability of a large range of bacteria to leach REEs that I would have thought more effective microorganism could have been chosen.

Prof Elizabeth Watkin

Space station biomining experiment demonstrates rare earth element extraction in microgravity and Mars gravity

RESPONSE TO REVIEWERS' COMMENTS

Reviewer 1

This is a very interesting report of an investigation of the potential for biomining to be employed in an extraterrestrial environment. Here, the authors employ both a flight-based and accompanied by a ground-based investigation of several representative cultures to extract rare Earth elements (REEs) under microgravity and partial gravity conditions. Given that a number of REEs are associated with meteorites and other extraterrestrial environments, a proof-of-concept study is certainly relevant. Here, the authors employed three test organisms, that were chosen for their abilities to withstand desiccation, form biofilms, grow on a common test medium, and interact with the rock surfaces. The authors provide a very detailed description of the various experimental conditions encountered during the flight, which is highly relevant to understanding the challenges involved. Overall, the data does support the proof of concept. I have some minor considerations that I would like the authors to consider.

Thank you for these positive comments. We provide a point-by-point response to the individual comments below:

1. In lines 92-94, the authors claim that prokaryotes are unable to directly sense gravity. There are a number of published experiments that would counter this claim (e.g. work by CA Nickerson, A Matin, and others).

Thank you for this statement. We have tempered this sentence to say: 'Although the capacity of prokaryotes to directly sense gravity remains a point of discussion' (lines 94-96). We want to avoid the introduction getting too far into this point as it can be a major distraction from the main message in the paper, but we agree that the sentence is best left open.

2. In line 485, the authors mention that dry heat sterilization did not alter the mineralogy of the basaltic rocks. While likely a very minor concern, what data supports this claim?

This is a helpful point to raise. This is the case because basalt contains no hydrous minerals. As long as the rock is kept dry when being sterilized, it will not undergo alteration or oxidation (minerals melt at temperatures far higher than an autoclave, i.e. >1000°C). We also know this because we routinely examine the mineralogy of these rocks by XRD and the treatment does not alter them. However, we have added in: 'as determined by X-Ray Diffraction (XRD)' to clarify this (line 490). We should say that the fact that this treatment would not alter a basalt rock should be clear to geologists.

3. The authors are to be commended for the very thorough characterization of the experimental issues (contamination, gravity variance due to the ISS-based centrifuge rotor size, temperature fluctuations etc.) that occurred during the flight experimentation.

Thank you for this positive review.

4. While the work presented supports the proof-of-concept of microbial mineral recovery of

rare Earth elements (REE) in extraterrestrial environments, the experimentation as presented represents several organisms grown as monocultures. Realistically a biomining setup would involve some type of polymicrobial association. In this context, some of the contaminated samples originally not included in this analysis, might provide an anecdotal insight into this possibility, and as a suggestion the authors may consider including some of this data in the supplemental information. Obviously, a full investigation would entail an analysis of the organisms that are present (i.e. metagenomic analysis) which would be outside the full scope of the reported work. This comment is intended as a suggestion.

This is a good point. However, the fixative prevented us from getting sufficient quality DNA to identify the contaminant. We should also point out that this was just one sample, so any inferences about a mixed microbial community would not be statistically meaningful even if we did have a reliable ID on the organism.

Reviewer 2

The manuscript explored bioleaching of REE under microgravity and Earth gravity conditions at International Space Station and on Earth. The results are novel and of interest to both space and biomining communities.

Thank you for this positive review of the work.

In a number of places the statements of the authors need clarification on the context to make it more clear to the reader what they are referring to.

We have responded to each of the comments below.

The solubilised REE concentrations are very low (about 1000 times lower than reported in some previous REE bioleaching studies). The authors should compare the leached concentrations to the concentrations reported to be bioleaching in previous studies from various REE minerals. The authors should also comment on whether such low concentrations as reported in this study would really be economical to recover using the proposed bioleaching approach.

We do discuss this point in the manuscript, where we state (lines 409-416):

'Our experiment has several differences with any proposed large-scale biomining activity. The basalt rock was not crushed in order test for biofilm formation on a contiguous but porous rock surface, another main goal of the BioRock experiment. This may have influenced the total percentage of REEs extracted from the rock, which was generally less than 1×10^{-5} %. These leaching rates would likely be higher with crushed rocks, which on Earth have been shown to result in leaching efficiencies of REEs of 8.0×10^{-3} % to several tens of percent under optimised conditions⁵⁰⁻⁵¹, which could be more representative of optimised commercial operations.'

This paper is a description of a scientific experiment, not an optimally designed commercial bioreactor set-up. We don't think it is useful to make economic statements here because that would involve taking our numbers and assessing the amount of REEs in a given volume of rock, whether you could sell that material at a market price etc. and these are complex economic calculations that are probably not helpful in a primary research paper.

Throughout the manuscript, please do not use active expressions such as “We investigated”, “we found”, “our experiment” but rather use passive expressions “was investigated”, “was found”, “the experiment” instead.

We have gone through the manuscript and make changes in most locations where we have used ‘we’ apart from in one place in the abstract (as this seems consistent with Nature Communications abstracts to state explicitly what ‘we’ found) and in a couple of places where it seems appropriate e.g. ‘we hypothesised’.

References, please provide all authors for each of the references in the reference list, instead of saying et al.

This formatting is in line with editorial requirements in Nature Communications.

Specific comments

Lines 34 and 70 and 103 and throughout the manuscript, please use small initial letters for “rare earth elements”

That has been edited.

Line 61, “accelerates” compared to what? Please specify

We have changed this to ‘Microorganisms can catalyse’ for clarity (line 59).

Lines 61-63, The sentence is too simplistic. Please note that commercial operations which utilise microbes for biooxidation of refractory gold ores and concentrates as a pre-treatment still typically rely on use of cyanide for gold solubilisation. Hence in the case of refractory gold ores and concentrates, the use of microbes does not typically avoid cyanide use. Moreover, copper is not typically extracted with cyanide. Rather traditional processes have been flotation followed by smelting and chemical leaching. Please revise the text accordingly.

Thank you for this comment. We have edited this to: ‘This process can in some circumstances reduce the environmentally damaging use of toxic compounds such as cyanides^{9,10}.’ (line 60-62). This seems to satisfy the fact that it may still be used in one part of the process, e.g. with gold, and it is not always the case that biomining avoids cyanide where it was never used in the first place, e.g. copper.

Line 63, which “mechanism” are the authors referring to? Several different mechanisms can be used to bioleach metals, e.g. acidolysis, redoxolysis, complexolysis, or e.g. contact leaching, non-contact leaching and cooperative leaching.

This has been clarified to: ‘These microbial interactions with minerals are used to decontaminate polluted soils’ (line 62-63).

Line 64, please add “in” before “a process called”

This has been added.

Lines 65-68, the statements are somewhat conflicting. If the bioleaching by heterotrops relies

on proton or organic acid production and decrease of pH, how can the heterotrophs be effective in bioleaching at circumneutral pH?

Thank you for noting this. This is referring to the point that the bulk pH can be circumneutral, but the microbes can change the pH locally to more acidic conditions to allow for leaching. We have added in the word 'local' to clarify this, i.e. 'These organisms can enable leaching by changing the local pH in the environment' (line 66-67).

Lines 103 and 108, please replace “Rare Earth Elements (REEs)” with ‘REEs’ as the term has already been previously defined.

This has been removed.

Line 104, please remove “and biomining” as it is redundant when bioleaching has already been mentioned

We have removed 'bioleaching' here to avoid redundancy and kept in biomining (as that is the word used in the abstract).

Lines 108-109, the statement is too simplistic as based on abstract “Bacillus subtilis” did not enhance leaching as compared to abiotic control although “Sphingomonas desiccabilis” did and no difference was observed for “Cupriavidus metallidurans”. In fact the whole paragraph does not mention which of the three species the results refer to. Please amend the paragraph to specify what species they refer to.

We have changed the heading (also in line with Nature Communications editorial policies to reduce character count) to 'Rare earth element (REE) biomining in space' (line 111). We have then ensured that throughout this section it is absolutely clear the beginning of each paragraph which organism we are referring to.

Figure 1b, please clarify what the yellow colour in the figure indicates

The yellow is the membrane. We have added into the parenthesis '(shown here in yellow)'.

Table 1, Please replace “concentration” with “content” (in caption and table) when talking about REE content in solids such as basalt, as the word concentration is typically used for solutions. It is also confusing that the authors say concentration (ng in chamber fluid volume of 6 mL). Do the values represent REE “mass” in 6 mL or have the authors calculated a concentration in the unit of ng/mL? Suggest using the latter to avoid confusion and adding the unit also into the table to improve clarity.

We have replaced 'concentration' with 'content' for the basalt column description. We have clarified the REE concentration as 'the total nanograms leached into the chamber fluid volume of 6 mL'. We prefer to keep this number rather than ng/mL because it shows the total mass of REE that was leached in the reactor. If it is expressed as ng/mL it implies one could predict how much one would get in a different volume by simply scaling it, but this is not necessarily the case. The total mass leached is likely strongly tied to the chamber volume as this determines the chemistry around the rock. From a biomining point of view, the number we currently have is a direct statement of how much REE we 'mined' from our rock slice into the chamber in this specific experiment.

Figure 2, the horizontal axis title and values are too small to read. Please increase font size to improve readability

This has been improved for clarity with both values and horizontal titles increased in font size.

Figure 2, the figure has text “ground experiment”, but does not show which data is from “experimental” flight”. Please clarify the distinction in the figure

This has been edited for clarity with ‘ISS’ added on the vertical axis to show clearly which are the flight experiment values.

Line 162, the authors should clarify what the percent values refer to, e.g. “relative % difference in mean of biological experiments as compared to controls” as now the meaning is not clear.

This sentence has been clarified to: ‘For S. desiccabilis, across all individual REEs and across all three gravity conditions on the ISS, the organism had leached 111.9 % to 429.2 % of the non-biological controls (Fig. 3a and Supplementary Table 3).’ (line 167-169).

Figure 3, please use thicker lines for the figure borders (e.g. axis), similar to those used for the legend box.

This has been edited for clarity to increase border thickness.

Supplementary table 2, please clarify in table caption whether the REE quantities are for soluble or total REEs. Please add headings for the first two columns and indicate in table which experiments were done at ISS.

This has been edited for clarity on all points. Headings have been added to the two columns, and ‘on ISS’ has been added to make the flight experiments completely clear.

Supplementary table 3, Please add headings for the first two columns and indicate in table which experiments were done at ISS

This has been edited for clarity on all points as above.

Supplementary table 4, please clarify to what data the differences refer to as it has not been specified in the caption. Also please indicate in the table which experiments were done at ISS

This has been edited for clarity on all points, in particular by making it clear that these comparisons are between conditions in Supplementary Table 2. As with the previous tables, ‘on ISS’ has been added to those conditions on ISS to make it absolutely clear.

Supplementary table 6, and indicate in table which experiments were done at ISS and clarify how the % was calculated. The current description “expressed as a percentage of total in the pellet and bulk chamber fluid (%)” is confusing. Do you mean “mass in pellet x100% / (mass in pellet + mass in bulk fluid)”?

This has been edited for clarity on all points as above. Concerning the percentage, we agree this could be clarified and we have now changed this to: 'Quantity of REEs in cell pellet as a percentage (%) of the total REEs in the pellet and bulk chamber fluid calculated from REE masses.

Supplementary table 7, please indicate in the table which experiments were done at ISS.

This has been added.

Lines 192-193, the statement is confusing. “difference was “significantly less” than what? Do the authors mean that bioleaching resulted in significantly lower REE solubilisation than control leaching both in microgravity and simulated Mars gravity, but under Earth gravity the bioleaching was not significantly different from that of the control? Please revise for clarity.

Yes, this is what this sentence means, and it has now been edited for clarity as follows: 'For B. subtilis, the quantity of REEs bioleached was significantly less than the non-biological controls in microgravity (ANOVA: $F(1,69) = 13.05, p < 0.001$) and simulated Mars gravity (ANOVA: $F(1,83) = 29.55, p < 0.0001$), but marginally not significant in Earth gravity (ANOVA: $F(1,83) = 3.79, p = 0.055$).' (line 198-201).

Lines 208-210, the sentence sounds like the authors would be referring to REE contents in the C. metallidurans cells and refer to Supplementary table 4, which does not state for what parameters the p values are shown. The authors should clarify the sentence to indicate that they refer to soluble concentrations instead of REEs in the cells to avoid confusion. Also Supplementary Table should clearly state from what parameters (e.g. soluble REE concentrations) the p values have been derived.

We have added in 'Comparisons were made for each REE leached into solution' in the main text to make it completely clear we are talking about leached REEs in the chamber solution, not the cells themselves (line 209). The caption to supplementary Table 4 has been edited to make clear the p tests for the comparisons refer to REEs in solution shown in Supplementary Table 2.

Lines 211-216, the authors should specify whether they talk about soluble REE concentrations or something else as it is not clear from the paragraphs as it currently stands

As for the comment above, we have added in 'leached into solution' for clarity.

Lines 217-219, a one sentence paragraph is too short to stand on its own. Suggest combining with the paragraph where the % leaching is given for the third species.

We have linked this to the paragraph above it to avoid it standing alone. It seems logical here as it summarises the amount leached as a percentage right after the discussion in the same paragraph of the statistics for the leaching of the individual REEs.

Line 221, please spell out abbreviation “ICP-MS” when first time mentioned

This is spelled out in Methods.

Lines 222-223, the authors state “The concentrations of REE in these samples generally

accounted for less than 5 % for all the elements in all the cultures on the ISS,” With “all the elements in all the cultures”, do the authors mean also elements such as carbon, nitrogen, hydrogen, oxygen, phosphorus? Or do the authors mean “% of REEs leached into culture medium”. Please clarify as the sentence is very confusing.

Thank you for pointing this out. We have clarified this sentence to: ‘The concentrations of REEs in these samples generally accounted for less than 5 % of the total REEs leached into solution in the biological experiments, with a few exceptions’ (line 29-230).

Lines 226-231, please clarify what differences the authors are referring to? % of REEs accumulated in cells? Or something else?

This sentence has been clarified to ‘ANOVA was used to ascertain whether the biological enhancement of REEs leached into solution exhibited by S. desiccabilis was also reflected in the quantity of REEs bound to cells.’ (line 232-234).

Line 232, what do the authors mean by pellet concentrations? Please clarify as the term is confusing. One could think that the authors are referring to cell numbers, or content of REEs in pellets, but I assume this is not the case. Do the authors mean the % of leached REEs that accumulated in cells?

Thank you for pointing this out. This has been clarified to: ‘but in almost all cases the percentage of total REEs associated with the S. desiccabilis cell pellets were lower than the other two organisms’ to make it clear it is the cell pellet (line 239-242).

Lines 234-235, what do the authors mean by “supernatant produced cell pellet preparation”? please clarify

This has been clarified to ‘The concentrations of REEs in the supernatant produced from washing of the cell pellet was below the detection limit.’ (line 242-244).

Lines 236-241, it is not clear what parameter the authors are discussing here in terms of differences in gravity conditions. Please clarify if the authors are comparing soluble REE concentrations or something else. Full stop is missing at the end of the paragraph.

This has been clarified to: ‘Comparison of the REEs leached into solution between the different gravity regimens of the non-biological control samples on the ISS’. A full stop has been added. (line 245)

Line 250, please specify which REEs are considered as heavy and which as light

This has been added in as follows: ‘S. desiccabilis caused preferential leaching of heavy REEs (Gd up to Lu) over light REEs (La up to Eu)’ (line 258-259).

Line 262, please replace “compated” with “compared”

We could not find this error here (we also did a search for it and was unable to find this typo or one similar to it). There is a ‘compared’ here, but it is correctly spelt.

Lines 267-269, please combine with previous paragraph, as one sentence paragraph is too short to stand on its own.

This has been combined with the previous paragraph.

Lines 270-282, please clarify if the authors are comparing soluble REE concentrations or something else as it is not clear

We have clarified this with: 'with the concentration of REEs leached into solution ...' (line 284).

Lines 280-282, please combine with previous paragraph, as one sentence paragraph is too short to stand on its own.

This paragraph has been joined with the one above it.

Line 218-282, what do the authors mean by “with simulated Earth gravity being higher”? Please clarify if the authors are referring to soluble REE concentrations being higher under simulated Earth gravity than ...

This has been clarified to: 'Non-biological controls exhibited a significant difference between the simulated Earth gravity on the ISS and ground controls (ANOVA: $F(1, 68) = 6.90, p = 0.011$) with the concentration of REEs leached into solution in simulated Earth gravity on the ISS being higher than ground controls across all REEs.' (line 281-283).

Supplementary Table 7, the authors should clarify in table caption whether the pH was measured after addition of the fixative which changed the pH to avoid misunderstandings.

This has been clarified in the caption to: 'Solutions were measured at the end of the experiment after sample fixation and return to Earth.' We should note that return to Earth automatically implies after fixation since fixation was done in space, but we agree that this edit makes it completely clear.

Lines 293-294, how do the authors explain the increase in pH in the presence of microbes. Previously they mentioned that the bioleaching with heterotrophs relies on production of protons and organic acids.

We do not know exactly what explains this increase in pH. It could be consumption of the R2A that produces a mildly alkaline pH. We agree that this might be seen to be at odds with the introduction. The introduction was an example of how heterotrophs can cause leaching by acidifying the medium, but we also discussed the release of complexing agents as another mechanism. In response to this comment, we have modified these sentences in the Introduction to make it absolutely clear that bioleaching can occur in a range of pH conditions, and we have exchanged one citation for a citation about alkaline bioleaching. It now reads: 'Acidophilic iron and sulfur-oxidisers are often used to biomine sulfidic ores, but heterotrophic microorganisms, including bacteria and fungi, can be effective in bioleaching in environments with circumneutral or alkaline pH¹¹⁻¹⁵. Organisms can enable leaching by changing the local pH in the environment, for example by the release of protons or organic

acids. Alternatively, leaching and sequestration of elements can occur as a consequence of the release of complexing compounds.’ (line 63-69).

We should say that we do not think that this small increase in pH is responsible for leaching in our experiments since it is observed in all three organisms and if this was responsible for leaching, we should expect to see leaching in all three organisms, which we do not.

Line 319, please abbreviate genus name as it has been defined before

All instances of genus have been checked for abbreviation after the first time.

Line 344, what do the authors mean by “enhancement of the heavier REEs”? please clarify whether they refer to enhancement of leaching of heavier REEs and as compared to what? it is not clear from the text as it stands

This has been edited for clarity to: ‘A greater biological enhancement in the leaching of heavy compared to light REEs was observed’ (line 346-347).

Lines 357-358, please clarify how biofilm formation was observed?

*This was observed by confocal microscopy. We have edited this sentence to: ‘We have observed *S. desiccabilis* by confocal microscopy to form biofilms on the surfaces and at the edges of cavities on the basalt more pervasively than *Bacillus subtilis* and *Cupriavidus metallidurans* under these growth conditions’ (line 360-362).*

Line 359, please abbreviate genus names as these should have been defined earlier

All instances of genus have been checked for abbreviation after the first time.

Line 421, please remove “faithful”

This has been removed and replaced by ‘reliable’ (line 421).

Line 435, please abbreviate genus name

All instances of genus have been checked for abbreviation after the first time.

Lines 455-456, please specify whether *Sphingomonas desiccabilis* is motile or not. In the discussion section, please discuss whether the mobility or non-mobility of cells plays a role on the possible impact of gravity on the cells and bioleaching.

*We have added that it is non-motile. We are reluctant to start discussing motility versus non-motility and effects on bioleaching mainly because the *S. desiccabilis* is non-motile in all conditions and so this is a factor that is controlled. It is true that the other two organisms can be motile, but we actually don’t know the exact motility state in space (clearly cells bound to the rock were not motile), and their motility is not (as far as anyone knows) a reason why they did not perform biomining. Discussing this will lead to a wide diversion that will not improve the interpretation of the results.*

Line 474, please add “, respectively” after “Table 1”.

This has been added (line 480).

Lines 487-490, what was the number of cells used for *S. desiccabilis*?

The number has been added to the text: 'An overnight culture of the strain was grown in R2A 100 % at 20-22 °C until reaching stationary phase (OD600 = 0.88±0.09; approximately 10⁹ colony forming units per mL).' (line 492).

Line 492, please add “spores” before “per slide”

This has been added (line 497).

Lines 494-506, please specify the number of cells per slide for *C. metallidurans*

This has been added to the text: 'Basalt slides, each containing approximately 10⁹ colony forming units per mL'. (line 505-506).

Line 542, please spell out abbreviation “PCB”

This has been spelled out (printed circuit board) (line 548).

Line 551, please specify temperature for refrigeration

This is specified in the storage conditions in the next section so we have added here 'as described below' (line 557).

Lines 582-584, were the samples not filtered before nitric acid analysis? How did the authors differentiate between REEs accumulated in cells and REEs that were soluble in bulk medium? Wouldn't acidifying the samples before cell removal result in some REE being resolubilised/desorbed from cells into bulk solution?

The samples were spun down to remove the pellet and thus separate the pellet from the bulk fluid. We differentiated the bulk from the pellet since the pellet was spun down as described and analysed separately. Adding acid before cell removal would result in some of the REEs potentially being removed from the cell pellet. In the discussion we do recognise that reduced pH during fixation might have unbound REEs, but we have now edited this to 'The reduced pH caused during fixation and sample preparation may have unbound any REEs attached to cell surfaces in all three species' to ensure that this point is covered.

Lines 595-596, the sentence does not read well as there is only “In” outside the parenthesis. Please revise for readability

This has been revised for clarity (line 600-602).

Line 622, please spell out abbreviation “RF”

This has been spelled out (radio-frequency) (line 627).

Line 642, poor English “and took into account”, please revise. Also please combine the

sentence with previous paragraph as one sentence is too short as a paragraph to stand on its own.

This has been edited to: 'Raw ICP-MS data (determined in µg/L) was converted to obtain the absolute quantity of a given element in the culture chamber, taking into account dilution factors applied during ICP-MS analysis.' (line 646-648).

Reviewer 3

This manuscript describes an experiment undertaken on the ISS investigating the bioleaching of REEs from basaltic rock under three different gravity conditions as well as a similar experiment on the ground at true Earth gravity. The data presented appears to be a subset of the full data set collected from the experiment (see ref #30).

While interesting, the leaching data on its own is not compelling enough for publication in Nature Communications. A more interesting story could be told by including the microbiological growth and biofilm formation under the same conditions.

We do not agree that this is the case. The growth data shows no difference between the gravity conditions and we cite that work in this paper, but presenting all that data separately here would not add to the bioleaching data or provide any additional explanatory power. However, in response to this comment, we have added the final cell concentration data as a Supplemental Table (Supplementary Table 9) so that the data in table format is directly accessible to the reader and we provide that link in the Methods where we talk about the final cell numbers.

We also do not think that biofilm data would add anything to the data and its interpretation. Adding in details about where the biofilms grew and their structure would take the manuscript on a large diversion, significantly increase the length of the paper and the data would not actually change the conclusions we present here. The reason for this is that we did not measure the REEs in the biofilms (because we needed to non-destructively study them) as stated in the paper. Therefore, data about the biofilms would not illuminate the biomining story by providing insight into the mechanism other than the obvious point that microbe-rock contact might influence leaching, which we already state. We might also note that the biofilms are localised around the edges of cavities (we state this in the paper). They are not a pervasive cell covering over the rock and for that reason we do not think that a detailed discussion about them is relevant to this paper which primarily focuses on the enhancement of REEs in the bulk fluid.

The amounts of REEs leached from the basaltic rock are minuscule at 0.00001% albeit significantly different from the abiotic control in the case of *Sphingomonas desiccabilis*. I query why the three particular microorganisms were chosen for this work. With the exception of a few REEs bioleached by *Sphingomonas desiccabilis*, leaching was not significantly different from the abiotic control even in the ground experiment. There have been many studies in the last 10 years investigating the ability of a large range of bacteria to leach REEs that I would have thought more effective microorganism could have been chosen.

We think that what are perceived to be negative results should not be seen in this way. As we discuss, microgravity has significant effects on organisms generally and perhaps we would find that although two of the organisms did not perform leaching in the ground experiment they would do under the static conditions in microgravity in which microbial stress is known

to be induced. Both the Bacillus and Cupriavidus species are known to interact with minerals and carry out transformations in metal-rich environments. Next to their scientific value, these species were chosen because of their compatibility with ISS safety. Our data, as we state in the paper, also show that ground based experiments give reliable insights into how we can expect organisms to behave in bioprocessing in space. That is a useful observation to make.

The overall low concentrations of REE leached is true. However, in our paper we raise this and we compare it to other bioleaching experiments, discussing why this is the case (for example, our rocks were not crushed as we wished to study biofilm growth as well). It is important to understand, as we state in the paper, that this is a scientific experiment to study microbial interactions with rocks and their applications to biomining, not a fully economically optimised biomining set-up.

REVIEWERS' COMMENTS

Reviewer #2 (Remarks to the Author):

Thank you for thoroughly addressing each of my comments and suggestions. The manuscript is now much more clear and easy to understand. I have no further suggestions for improvements.

Reviewer #3 (Remarks to the Author):

Thank you for addressing the majority of my concerns. It does need to be pointed out that it is perfectly possible to study biofilm formation on crushed rocks. For example please see the following paper

<https://www.mdpi.com/2075-163X/6/3/71/htm>

Elizabeth Watkin

Space station biomining experiment demonstrates rare earth element extraction in microgravity and Mars gravity

RESPONSE TO REVIEWERS' COMMENTS

Reviewer 2

Thank you for thoroughly addressing each of my comments and suggestions. The manuscript is now much more clear and easy to understand. I have no further suggestions for improvements.

Thank you for the positive assessment of our responses.

Reviewer 3

Thank you for addressing the majority of my concerns. It does need to be pointed out that it is perfectly possible to study biofilm formation on crushed rocks. For example please see the following paper

<https://www.mdpi.com/2075-163X/6/3/71/htm>

We accept that biofilm growth occurs on crushed rocks and thank the reviewer for the paper link. What we were emphasising here is that we wanted to study biofilm growth on a flat surface (crushed rocks make it very difficult to quantify biofilm growth if the fragments are irregular three-dimensional surfaces). We have clarified this to: 'The basalt rock was not crushed in order to investigate biofilm formation on a flat, contiguous but porous rock surface, another main goal of the BioRock experiment.' (*lines 404-405*). This should hopefully clarify this point, but we think that extending this discussion further to talk about the merits and de-merits of studying biofilms on fragments versus flat surfaces will be a distraction from the focus of the paper.